# Developing an In-House Comprehensive Medication Review Training Program for Clinical Pharmacists in a Finnish Hospital Pharmacy

**DOI:** 10.3390/ijerph20126158

**Published:** 2023-06-16

**Authors:** Kirsi Kvarnström, Ilona Niittynen, Sonja Kallio, Carita Lindén-Lahti, Marja Airaksinen, Lotta Schepel

**Affiliations:** 1HUS Pharmacy, Helsinki University Hospital and University of Helsinki, 00029 Helsinki, Finland; ilona.niittynen@hus.fi (I.N.); carita.linden-lahti@hus.fi (C.L.-L.); lotta.schepel@hus.fi (L.S.); 2HUS Internal Medicine and Rehabilitation, Helsinki University Hospital and University of Helsinki, 00029 Helsinki, Finland; 3Clinical Pharmacy Group, Division of Pharmacology and Pharmacotherapy, Faculty of Pharmacy, University of Helsinki, 00014 Helsinki, Finland; marja.airaksinen@helsinki.fi; 4The Association of Finnish Pharmacies, 00510 Helsinki, Finland; sonja.kallio@apteekkariliitto.fi; 5Quality and Patient Safety, Shared Group Services, Helsinki University Hospital and University of Helsinki, 00029 Helsinki, Finland

**Keywords:** clinical pharmacy, medication safety, pharmacy practice, continuing education, medication review, comprehensive medication review, medication therapy management

## Abstract

Long-term continuing education programs have been a key factor in shifting toward more patient-centered clinical pharmacy services. This narrative review aims to describe the development of Helsinki University Hospital (HUS) Pharmacy’s in-house Comprehensive Medication Review Training Program (CMRTP) and how it has impacted clinical pharmacy services in HUS. The CMRTP was developed during the years 2017–2020. The program focuses on developing the special skills and competencies needed in comprehensive medication reviews (CMRs), including interprofessional collaboration and pharmacotherapeutic knowledge. The program consists of two modules: (I) Pharmacist-Led Medication Reconciliation, and (II) CMR. The CMRTP includes teaching sessions, self-learning assignments, medication reconciliations, medication review cases, CMRs, a written final report, and a self-assessment of competence development. The one-year-long program is coordinated by a clinical teacher. The program is continuously developed based on the latest guidelines in evidence-based medicine and international benchmarking in cooperation with the University of Helsinki. With the CMRTP, we have adopted a more patient-centered role for our clinical pharmacists and remarkably expanded the services. This program may be benchmarked in other countries where the local education system does not cover clinical pharmacy competence well enough and in hospitals where the clinical pharmacy services are not yet very patient-oriented.

## 1. Introduction

Clinical pharmacy is an area of pharmacy dealing with the science and practice of the rational and appropriate use of medicines [1,2]. Clinical pharmacists are expected to ensure the quality and safety of medication therapies in patient care, emphasizing collaborative care and patient interaction [1,3,4,5,6]. Even though patient-centered clinical pharmacy services have been shown to improve the quality, safety, and efficiency of care and to reduce costs [7,8,9,10,11], their diffusion to many healthcare systems, e.g., in Europe, has been slow [12]. The European Directorate for the Quality of Medicines (EDQM) has invited European governments and policymakers to implement the principles and working methods of clinical pharmacy and pharmaceutical care in their national healthcare systems [13,14]. This has also been addressed in several Finnish guidelines and medicine policy documents during the last decades [15,16,17,18,19].

In Finland, pharmacists started to work in hospital wards in the 1980s [20]. Their tasks were very logistically and technically oriented for a long time, concentrating on dispensing, preparing, and storing drugs, aiming to reduce the nurses’ workload [20,21]. Implementation of international patient and medication safety initiatives started in Finland in 2005–2006, when the first national guidelines for medication safety were published [22,23]. This gradually changed the focus to more clinical and patient-oriented tasks. Along with long-term continuing education and research on medication-related risks, research on the benefits of clinical pharmacy services to manage these risks was needed to expand the role of clinical pharmacists in hospitals [24,25]. Around 2010, more patient-centered tasks started to become a more common part of pharmacists’ work in hospitals. During these years, interprofessional patient safety work combined with systems thinking started in Finland [20,26]. In addition to evolving clinical pharmacy continuing-training programs, this was clearly key in changing the role of hospital clinical pharmacists in Finland [25]. At the same time, logistical tasks were transferred to pharmacy assistants, which slowly increased the number of pharmacy assistants in hospital wards. This evolution also raised the demand for patient-centered clinical pharmacy services in Helsinki University Hospital (HUS, Figure 1) and motivated the appointment of new requirements for clinical pharmacists’ competence. In addition, the HUS Pharmacy renewed its strategy for clinical pharmacy services to one based on prioritizing the tasks of clinical pharmacists from a patient and medication safety perspective. In addition, the HUS Pharmacy initiated a strong collaboration with the University of Helsinki in medication safety research and practice development, which had a remarkable impact on this strategy work.

As the traditional role of Finnish pharmacists was in drug logistics and preparing medications, clinical pharmacy contents were, for a long time, missing from undergraduate pharmacy studies in Finnish universities. Finland has a two-tier university education program for pharmacists, consisting of BSc and MSc degrees. The three-year-long BSc (Pharm) degree consists of 180 ECTS credits, and the MSc (Pharm) degree, consisting of a total of 300 ECTS credits, takes an additional two years to complete [27]. The undergraduate pharmacy curriculum has been focused on the competencies required in drug research and development and community pharmacy practice, while it has not provided the competencies required to conduct clinical pharmacy services in hospitals. Community pharmacists’ involvement in medication counseling has been allowed and mandated by law since the beginning of the 1980s; therefore, the first features of clinical pharmacy competencies incorporated in the curriculum have been concentrated on providing medication counseling in pharmacies [28,29,30,31,32]. Because of the curriculum’s missing clinical hospital pharmacy contents, clinical pharmacists (pharmacists providing clinical pharmacy services, e.g., in hospital wards) have had to specialize in tasks of their own area alongside working without any mandatory additional education or residency program after the BSc or MSc studies.

Clinical pharmacy continuing-education programs can be seen as one key factor in shifting toward more patient-centered, clinical-pharmacy-oriented services in Finland [25,33]. The first step was long-term accreditation training for collaborative comprehensive medication reviews (CMRs) that was nationally initiated in Finland in 2005 [34]. This can be seen as a milestone in Finnish clinical pharmacy education and in the competence development of practicing pharmacists. More hospital-specific clinically oriented continuing education became available in 2010 when the national specialization program in hospital and health center pharmacy was started [20,35]. In particular, competence in conducting medication reviews can be seen as a core skill in conducting clinical pharmacy services [3,12,36,37,38,39]. This was not covered by undergraduate pharmacy education in Finland before 2014, even though principles of pharmaceutical care and medication risk management have been systematically included in the curriculum since the beginning of the 2000s [40,41].

The increased demand for patient-centered clinical pharmacy services in secondary and tertiary care and the need to precisely obtain the special expertise for the HUS Pharmacy professional setting (university teaching hospital and an increasing number of other hospitals and healthcare units) led to the idea of developing a tailored in-house training program for comprehensive medication reviews. The goal was to provide clinical pharmacists with the required knowledge and competence to work in care teams in specialized care. This review aims to describe the development of HUS Pharmacy’s in-house Comprehensive Medication Review Training Program (CMRTP) and how it has developed our clinical pharmacy services.

## 2. Developments in Finnish Pharmacy Education towards Clinical Pharmacy Competences

Pharmacy education in many countries has shifted toward meeting patient-care-oriented competence needs [42,43,44]. Acquiring medication review competencies via continued training alongside work may be time-consuming and too heavy a workload; thus, a need for acquiring similar skills as a part of Finnish undergraduate pharmacy education was identified by the University of Helsinki. The University of Helsinki changed its competency requirements and adopted medication reviews as a new competence area in its curriculum reform in 2014 [41,45]. Achieving medication review expertise is now built into the entire BSc (Pharm) degree program. It consists of theoretical studies and practical training as part of an internship that students complete in either a community pharmacy or partly in a hospital pharmacy [40,46,47].

Achieving medication review expertise has been illustrated using a pyramid model (Figure 2). Pharmacotherapy competence develops from a basic knowledge of medicines and medication counseling to medication reviews and CMRs [45,46,48]. The higher one goes in the pyramid, the greater the importance of interprofessional collaboration. After basic education, lifelong learning continues, and competence is then supplemented with continued education and skills acquired in working life.

## 3. Setting

Helsinki University Hospital (HUS) is the largest provider of specialized healthcare in Finland, providing secondary and tertiary care in 23 hospitals with approximately 3000 beds. HUS serves a regional population of 1.6 million in the capital area of Finland. The HUS Pharmacy provides medication management and clinical pharmacy services to HUS’s 24 member municipalities, Kymsote (serving 170,000 residents in South-Eastern Finland), and the Finnish Defense Forces. The HUS Pharmacy provides medication management and clinical pharmacy services to primary, secondary, and tertiary care units, as well as for several social care units in the municipalities.

The HUS Pharmacy has provided clinical pharmacy services since the 1990s (Figure 1), and the tasks have gradually changed from drug logistics and technical tasks to more patient-centered tasks [20,21,25]. In 2016, when the development of the in-house training program was initiated, the tasks included ordering medicines for the wards, dispensing oral drugs, checking possible interactions, preparing parenteral medications, as well as providing medicine information for the ward personnel, and, in some units, also for patients and patients’ relatives. Piloting medication reconciliation and medication reviews had just started as part of the research in the emergency department [20]. At the same time, the number of clinical pharmacists started to increase rapidly because of the hospital’s interprofessional system-based patient and medication safety work where the HUS Pharmacy took increasingly more part.

## 4. The In-House Comprehensive Medication Review Training Program (CMRTP)

### 4.1. Piloting an In-House Clinical Pharmacy Training Program in 2016

The University of Helsinki and HUS Pharmacy received two years of funding to pilot a position for the first mutual clinical teacher (a university teacher with a Ph.D. (Pharm) planning, implementing, and developing hospital clinical pharmacy education) in 2014–2016. During these years, the clinical teacher monitored the need for skills to provide more patient-oriented clinical pharmacy services and developed a year-long pilot in-house training program for five clinical pharmacists. The program consisted of medicine information, high-alert medications, common adverse drug reactions, interpretation of laboratory test values, medication reconciliation, and one to two medication reviews. Unfortunately, the funding for the position of a clinical teacher was discontinued after two years. Senior pharmacists who had graduated from the national CMR accreditation program (comprehensive medication review) [34] continued further development of the in-house training program during the funding break because the training was perceived as crucial to clinical pharmacy services. At the same time, the CMR competence areas for pharmacists were nationally identified and incorporated into the undergraduate pharmacy curriculum in Finland [40,45,46,47,49]. Due to this, we decided to concentrate on CMR skills. These were also seen as core skills in conducting clinical pharmacy services.

### 4.2. The First In-House Comprehensive Medication Review Training Program (CMRTP) in 2017

The first in-house Comprehensive Medication Review Training Program (CMRTP) was held from January 2017 to December 2017 for ten clinical pharmacists working in HUS Pharmacy (Figure 3). It consisted of ten four-hour teaching sessions, self-learning (e.g., learning about local current care guidelines), five medication reviews, five comprehensive medication reviews (including patient interviews), a medication safety audit in the participating pharmacists’ working units, and a written final work about integrating medication reviews in their work tasks alongside a self-assessment of their competence development.

The teaching sessions included expert lectures, e.g., about drug interactions, interpreting laboratory results, renal and hepatic insufficiency and their impact on drug use, the impact of aging and drug use in older patients, drug use in several chronic diseases (e.g., cardiovascular and memory diseases, pain, insomnia, depression), and perioperative care. The use of electronic databases was also included. In the beginning, the lecturers were mainly specialized physicians and a senior pharmacist working in HUS since there was a lack of clinical pharmacists with deep knowledge of the topics. The three HUS Pharmacy’s comprehensive medication review specialists acted as tutors. The costs of this program were estimated to be approximately EUR 1500 per participant (including the working time of the coordinating senior pharmacist and rewards for the lecturers), while the cost of commercial education was approximately EUR 4000 per participant. The CMRTP was free of charge for the participating pharmacists, with 50% of the education conducted during their working time and 50% conducted during their free time.

### 4.3. Improving and Adjusting the CMRTP during 2018–2020

Funding for a clinical teacher was received again in 2017 by HUS and the University of Helsinki. The clinical teacher held the next one-year in-house training program from August 2018 to May 2019 for 15 clinical pharmacists. It was planned and built based on the previous year’s program, focusing even more on developing the knowledge and competence needed in CMRs (Figure 2). Medication review training was more systematic, as students made a detailed plan beforehand with their named partner physician for conducting the CMRs, and the conducted CMRs were retrospectively reviewed and approved by the tutors. These steps were added to the procedure to ensure that the CMRs were properly conducted, as some students from the previous CMRTP had difficulties understanding the CMR assignments correctly.

The third course for the in-house CMRTP was held from August 2019 to May 2020 for ten clinical pharmacists. As patient care and medication management are diverse in secondary and tertiary care and further differ from those in primary care, tailoring the program according to the specialty field or the type of the working unit was seen as essential. A remarkable update to the course was the orientation of each in-house CMRTP student to their area of specialization based on the unit in which they work. The specialization included participation in meetings intended for specializing physicians and familiarizing oneself with the literature of the specialty fields. This tailoring was not included in the commercial accreditation CMR training program or not widely achievable from clinical pharmacy continuing education programs, though the first residency program pilot started in 2018 for a few pharmacists attending specialization studies in the community and hospital pharmacy program at the University of Helsinki. The medication safety audit was removed from the program in 2019 since it was seen as a separate competence area that could be learned outside the CMRTP.

The tutoring process included new tutors, as more pharmacists had graduated from previous in-house programs. The tutoring procedures were standardized to ensure equal requirements and support offered for the pharmacists conducting CMRs and consistent quality of the knowledge and skills developed by the pharmacists.

### 4.4. Current Content of the In-House CMRTP

The core of the in-house CMRTP has remained the same since 2021 and provides for 15 clinical pharmacists per year; however, the current in-house CMRTP consists of two modules: Part (I) Pharmacist-Led Medication Reconciliation (MedRec), and Part (II) Comprehensive Medication Review (Figure 2 and Table 1). The pharmacist-led medication reconciliation was integrated into CMRTP in 2020 because the implementation of a new electronic patient record system (EPIC-based APOTTI) increased the need for pharmacist-led medication reconciliation in HUS significantly [50]. Integrating these two courses made it possible to train more pharmacists simultaneously. All pharmacists who complete the training program will be able to conduct MedRec and, by completing both parts, they will be qualified to conduct CMRs. The first MedRec course participants graduated in 2021.

Some updates have been made, e.g., regarding the contents and the tutoring process. As pharmacists in the HUS Pharmacy are working more and more extensively in specialized and primary healthcare, e.g., home care, the update from 2019 of the specialization orientation of each CMRTP student is still seen as essential. This enables them to develop their clinical pharmacy skills and competencies (Figure 3). Over the years, the in-house training program has increased the number of pharmacists with CMR competencies in HUS, enabling personal tutors for each student during their training. The program is continuously developed based on the latest guidelines in evidence-based medicine, international benchmarking, participants’ feedback, research, and other emerging needs.

### 4.5. Defining Adequate Comprehensive Medication Review Skills and Ensuring Competence and Ensuring Quality of the Training Program

In addition to the national CMR accreditation training programs, more options for acquiring medication review competencies and accreditation via studying besides work have been available in Finland in recent years. In 2017, national competence criteria for pharmacist-led collaborative medication reviews were established in Finland [49]. The recommendations are based on guidelines, legislation, and national and international research on CMRs. The competence criteria for conducting CMRs in HUS Pharmacy were adapted on the basis of these national competence recommendations, taking also into account the special competence needs of secondary and tertiary care. In October 2019, the HUS Pharmacy CRMTP was evaluated in the AATE group (The National Coordination Group of Professional Development of Pharmacy Services) and was considered to fulfill the national criteria.

Nowadays in Finland, there are several external commercial medication review education programs, and medication review competence is built into undergraduate pharmacy education (BSc Pharm). To ensure that pharmacists working in HUS Pharmacy have sufficient clinical pharmacy skills and competence to conduct CMRs in the highly specialized care setting, the HUS Pharmacy assesses their competence before permitting them to conduct medication reviews independently. When a pharmacist beginning work in the HUS Pharmacy has been trained for medication reviews outside the HUS pharmacy, they must successfully conduct two CMRs under the supervision of an experienced pharmacist as evidence of their competence. Through this process, we can ensure the quality and safety of CMRs conducted by clinical pharmacists.

## 5. Discussion

### 5.1. Evolution of HUS Pharmacy’s Clinical Pharmacy Services

During recent years, the provision of clinical pharmacy services in HUS Pharmacy has increased remarkably (Figure 1). The CMRTP has enabled a change in the role of clinical pharmacists from logistical and technical tasks to patient-centered services. Benchmarking international clinical pharmacy operating models, particularly in Europe and the US, has ensured that in-house training corresponds better to international medication management recommendations [37,38,39]. Medication reviews have become a part of several pharmacists’ work tasks, and pharmacist-led medication reconciliation has increased strongly. With the skills gained in Part I (i.e., the MedRec part) of the training program, our clinical pharmacists can better identify medication-related problems such as adverse drug reactions, drug interactions, and unnecessary polypharmacy [51]. Part II of the program extends their competence to a more comprehensive level, including, e.g., untreated indications, drug use without indications, interpreting laboratory values, drug use during aging, renal/hepatic diseases, and medications in their own area of specialty. The new EHR system, APOTTI, has promoted pharmacist-led medication reconciliation in several new emergency departments, outpatient clinics, and municipal health centers [50]. Clinical pharmacists have begun to offer independent appointments for patients where they can receive advice on their medication, medication reconciliations, and medication reviews. Pharmacists update patients’ home medication lists before physician appointments, which seems to reduce the physicians’ workload [51]. As a new task, pharmacists have become involved in the prescription renewal process by reviewing the appropriateness of the prescription before physicians renew [52].

As a part of patient safety improvement in HUS, medication order verification conducted by pharmacists was piloted and implemented as a new clinical pharmacy service starting in 2019 [39,53,54]. This development of clinical pharmacy work tasks further raised the pharmacists’ competency and educational requirements to include the necessary knowledge and skills to decrease the risk of medication errors in the physicians’ orders. In medication order verification, a clinical pharmacist reviews and validates the physicians’ orders as soon as possible, preferably within 24 h of the order being given. This is a new task for clinical pharmacists in Finland, and with the APOTTI EHR system, we have a feasible technical workflow for this process [50].

### 5.2. Lessons Learned and Future Plans in Developing Clinical Pharmacy Services and the CMRTP

The future aim of HUS Pharmacy is to train all our clinical pharmacists to perform comprehensive medication reviews (the highest stage in the learning pyramid, Figure 3). With these skills, the principles of pharmaceutical care can be implemented in their mindset and working methods. Since pharmacy residency programs allowing an individual choice of specialization area are not widely achievable in Finland, it is valuable that the in-house CMRTP enables tailoring. The tailoring of the CMR concepts to each pharmacist’s working environment helps them to become a clinical specialist in the field of specialty in which they are working. Due to this, standard interprofessional teamwork practices are easier to acquire and maintain. In Finland, it was commonly thought that CMRs are not applicable to operating rooms or intensive care units, among other settings, but we have now learned to apply these skills to these environments as well; however, interventions based on CMRs can be time-consuming, and not all medication changes can be implemented in tertiary care. More efforts should be made to inform primary care professionals about the findings of CMRs, hopefully electronically in the near future.

In addition to conducting medication reviews, CMR skills can be used in medication reconciliation and order verification processes. These tasks are rapidly expanding, even though the latter is new for clinical pharmacists in Finland. HUS is the first Finnish hospital in implementing a closed-loop medication management process. Order verification is a crucial safety defense because barcode scanning and automated dispensing systems depend on correct orders [50]. In the HUS Pharmacy, we have started verifying physicians’ orders with the clinical pharmacists working during weekday daytime shifts. The Joint Commission International (JCI) recommends order verification within 24 h after ordering medication [53]. In the future, we are hoping to have pharmacists also verify medication orders on weekends, in which case the JCI standards will be fulfilled.

Training all clinical pharmacists to perform CMRs is possible only if we have a clinical teacher developing and coordinating the program. Hence, permanent funding for the clinical teacher’s position is crucial. The importance of peer support also turned out to be vital, as we have more CMR-trained tutors from several specialties now available. In the best case, the tutor works in the same specialty with a similar patient profile and interprofessional team members. This type of learning method—one that includes interprofessional collaboration and learning from other specialists—leads to a more efficient learning process while residency programs are not yet widely available in Finland. The participation of doctors as trainers is also essential from the perspective of strengthening collaboration and creating standard practices.

Providing the CMRTP has helped us to recruit pharmacists, as many of them have said that the CMRTP was one of the reasons for applying to work at the HUS Pharmacy. Hospital pharmacies all over Finland have been interested in HUS Pharmacy’s in-house CMR training and have been willing to benchmark it. Our program may also be benchmarked in other countries where the local education system does not cover clinical pharmacy skills well enough or is not reachable for high volumes of participants. Furthermore, it can be benchmarked in hospitals where the clinical pharmacy services are not yet very patient-oriented. We hope that this narrative review may help those who plan to build an in-house comprehensive medication review training program to ensure the quality and safety of medication therapies in patient care.

## 6. Conclusions

With the comprehensive medication review in-house program (CMRTP), we have adopted a more patient-centered role for our clinical pharmacists and remarkably expanded our services relatively quickly. We have also achieved competence suitable for our needs, especially for secondary and tertiary care. Our program may be benchmarked in other countries where the local education system does not cover clinical pharmacy skills or in hospitals where the clinical pharmacy services are not yet very patient-oriented. Implementing a new EHR system has also sped up this change, and with the CMRTP, we were well prepared for the increased supply.

## Figures and Tables

**Figure 1 ijerph-20-06158-f001:**
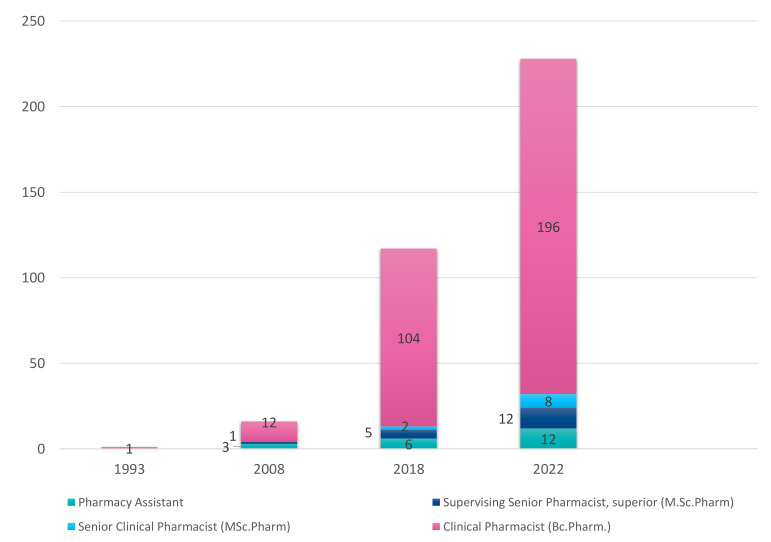
Numbers of pharmacists and pharmacy assistants working with clinical pharmacy services in HUS Pharmacy in 1993, 2008, 2018, and 2022 (HUS Pharmacy’s internal data).

**Figure 2 ijerph-20-06158-f002:**
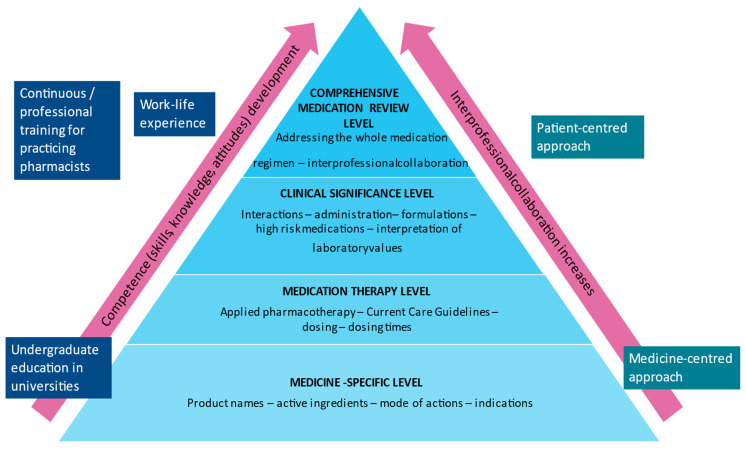
The learning pyramid illustrates the development of pharmacists’ patient care and medication review competence during undergraduate education and the continuous competence development through professional training and work experience [modified from [45,46,48]].

**Figure 3 ijerph-20-06158-f003:**
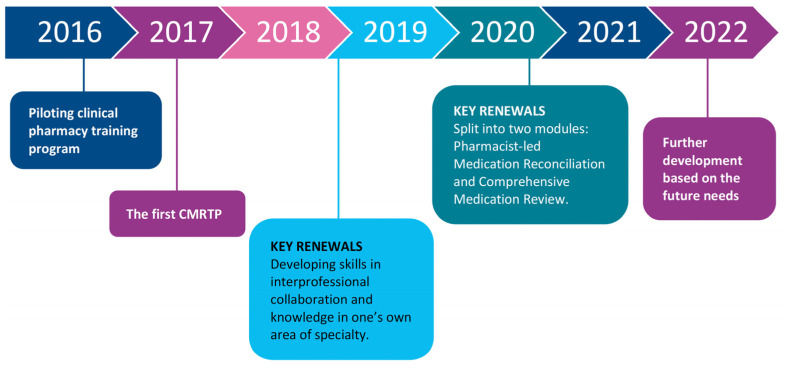
Key development steps in the in-house Comprehensive Medication Review Training Program (CMRTP) for clinical pharmacists in HUS Pharmacy.

**Table 1 ijerph-20-06158-t001:** Current content of the in-house Comprehensive Medication Review Training Program (CMRTP) for clinical pharmacists in HUS Pharmacy.

Credits (ECTS *)	Twenty, Split into Two Modules: Pharmacist-Led Medication Reconciliation (Part I) and Comprehensive Medication Review (Part II).
Lectures and workshops	Part I: Four four-hour lessons Topics: Drug interactions; Use of drug information databases supporting medication reviews; Identifying medication-related problems and assessments of clinical significance; Patient interviews; Medicine information and patient counseling on a patient ward; Medication adherence; Drug use in several diseases (e.g., cardiovascular and memory diseases, pain, insomnia, depression) and perioperative care; Pharmacist-led medication reconciliation in practice.
	Part II: Six four-hour lessons Topics: Interpreting laboratory results; Renal and hepatic insufficiency and drug use; Impact of aging and drug use in older patients, PIMs for older adults, deprescribing; Interprofessional collaboration in medication reviews; Comprehensive medication reviews in practice, how to identify patients in need of a CMR.
Assignments	Personal learning plan; Learning diary; Reading the literature and professional meetings related to one’s own area of specialty; Self-learning between the live sessions (e.g., current care guidelines); Final reflective report: integrating comprehensive medication reviews in own work tasks.
Case-based medication review training	Four pharmacist-led medication reconciliations; Three medication review cases based on patient scenarios; Five comprehensive medication reviews for real patients.

* ECTS = The European Credit Transfer and Accumulation System; PIM = Potentially Inappropriate Medication; CMR = comprehensive medication review.

## Data Availability

The data used in this study are available on request from the corresponding author.

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
