# Peer review of "Developing an In-House Comprehensive Medication Review Training Program for Clinical Pharmacists in a Finnish Hospital Pharmacy"

_ijerph, 2023, doi:10.3390/ijerph20126158_

Round 1

Reviewer 1 Report

This is a very interesting work, which moreover deals with a theme that has probably not been discussed enough to date. The article is well structured and clear; only a few remarks should be brought to the authors' attention.

- Title: since only the case of HUS is discussed in this review, the Finnish context of this work should be mentioned in the title.

- line 17: please develop the HUS abbreviation.

- lines 51-53: this sentence is unclear, please reword.

- lines 59-60: this sentence is unclear, please reword.

- lines 103-104: maybe “implemented” would be more appropriate than “developed”.

- Paragraph 4.2: specifying the overall time period (one full year? Several months?) over which this first CMRTP took place would be interesting.

- line 224: please remove the coma between “MedRec” and “and”.

- Discussion part: comparisons with other countries regarding these clinical pharmacy activities would be of great value in this part.

English language is good; minor editing of English language required

Reviewer 2 Report

Dear Colleagues

It was a pleasure to read your work and I would like to congratulate you on your initiative.

I would like to make some comments so that you can improve your manuscript:

- Firstly, I would remove from the title - narrative review - since your text is not in line with this methodology. In methodological terms it would be something more like action-research;

- please put the source of the images or reference of the original data

- Figure one, the reader does not understand the difference between each type of pharmacist.

- It is important that you insert quantitative data about the participants in each edition of the course, to be able to analyse the evolution and diversity of each new edition.

Reviewer 3 Report

This is a good review describing the structure and the implementation of the comprehensive medication review program for pharmacists in Finland and its impact. A few comments are listed below for the atuhtors to consider.

1. Lines 70-80: In this paragraph, the authors described the academic pharmacy education in Finland. However, it is unclear what kind of additional education/training to those pharmacists have to get in order to be classified as clinical pharmacists. For example, in the US, graduates of pharmacy programs have to complete pharmacy residency, whereas in the UK, they have to complete the MPharm degree. Is the MSc degree described by the authors is what makes a pharmacist a clinical pharmacist? Please add more details.

2. Line 150: Please provide a definition for "clinical teacher" in between parentheses. Do you mean clinical pharmacist? Or what kind of profession did those clinical teachers have?

3. Line 179-180: Please justify why were all lectures physicians? Why none of them was a pharmacist? I noticed that after the graduation of some students from the program, they became tutors in this program (likes 212-215). If the reason for the lack of pharmacist lecturers was the lack of trained pharmacists on this topic, then this should be clarified.

4. Table 1: Remove ", a lecture and a workshop" next "Patient interview" since the topic is already listed in the row of "Lectures and workshops"

5. Table 1: Was Part II also lectures and workshops? If so, either merge it with the previous row (while keeping the side title of Part II) or type "Lectures and workshops" inside the cell to the left to be similar to Part I.

6. Line 250: Please spell out "AATE"

7. I wonder why didn't the program developers consider recruiting clinical pharmacists from countries with established clinical pharmacy services to teach in the program when funding was available back in 2016-2017? This probably could have been a limitation at the beginning of the program as having clinical pharmacists teaching in the program could have been better than having physicians.

Round 2

Reviewer 3 Report

Thanks to the authors for addressing the comments. I have no further comments.